# Monocyte Infiltration and Differentiation in 3D Multicellular Spheroid Cancer Models

**DOI:** 10.3390/pathogens10080969

**Published:** 2021-07-30

**Authors:** Natasha Helleberg Madsen, Boye Schnack Nielsen, Son Ly Nhat, Søren Skov, Monika Gad, Jesper Larsen

**Affiliations:** 1Bioneer A/S, Kogle Allé 2, 2970 Hørsholm, Denmark; bsn@bioneer.dk (B.S.N.); nhatson@hotmail.com (S.L.N.); mga@bioneer.dk (M.G.); jla@bioneer.dk (J.L.); 2Department of Veterinary and Animal Sciences, University of Copenhagen, Ridebanevej 9, 1870 Frederiksberg C, Denmark; sosk@sund.ku.dk

**Keywords:** 3D cancer cell models, multicellular spheroids, tumor-associated macrophages, tumor microenvironment, drug screening, in vitro assay

## Abstract

Tumor-associated macrophages often correlate with tumor progression, and therapies targeting immune cells in tumors have emerged as promising treatments. To select effective therapies, we established an in vitro 3D multicellular spheroid model including cancer cells, fibroblasts, and monocytes. We analyzed monocyte infiltration and differentiation in spheroids generated from fibroblasts and either of the cancer cell lines MCF-7, HT-29, PANC-1, or MIA PaCa-2. Monocytes rapidly infiltrated spheroids and differentiated into mature macrophages with diverse phenotypes in a cancer cell line-dependent manner. MIA PaCa-2 spheroids polarized infiltrating monocytes to M2-like macrophages with high CD206 and CD14 expression, whereas monocytes polarized by MCF-7 spheroids displayed an M1-like phenotype. Monocytes in HT-29 and PANC-1 primarily obtained an M2-like phenotype but also showed upregulation of M1 markers. Analysis of the secretion of 43 soluble factors demonstrated that the cytokine profile between spheroid cultures differed considerably depending on the cancer cell line. Secretion of most of the cytokines increased upon the addition of monocytes resulting in a more inflammatory and pro-tumorigenic environment. These multicellular spheroids can be used to recapitulate the tumor microenvironment and the phenotype of tumor-associated macrophages in vitro and provide more realistic 3D cancer models allowing the in vitro screening of immunotherapeutic compounds.

## 1. Introduction

Crosstalk between different cells in the tumor microenvironment (TME) plays an important role in tumor growth and tumor-mediated immune suppression in vivo. The presence of infiltrating immune cells in tumors often correlates with tumor growth and progression [1,2,3,4]. Especially, tumor-associated macrophages (TAMs) are abundant in most types of malignant tumors and can contribute to multiple cancer hallmark capabilities [5,6]. However, the TAM compartment is both highly dynamic and tumor-specific. This heterogeneity reflects the ability of macrophages to acquire an entire spectrum of phenotypic, metabolic, and functional profiles, ranging from a pro-inflammatory to an anti-inflammatory state [7]. In particular, M1-like TAMs can produce reactive oxygen species, destroy malignant cells, and release pro-inflammatory cytokines, further stimulating anti-cancer immunity [1]. Nevertheless, cancer cells often acquire the ability to polarize infiltrating monocytes toward an M2-like state with immunosuppressive and pro-angiogenic features that promote tumor development and invasion [7,8,9,10]. Furthermore, TAMs can either antagonize, augment, or mediate the antitumor effects of different anti-cancer therapies [6].

As the presence of M2-like macrophages in tumors correlates with suppressed T cell responses, tumor progression, and poor prognosis in patients, therapies targeting macrophages in tumors are consequently tested in clinical trials [11,12]. Evaluating therapeutic compounds in conventional 2D models possibly allows the study of a compound’s repolarizing effect on macrophages outside a TME; however, the effect on TAM infiltration or repolarization within the complex TME cannot be studied properly. This likely accounts for the low success rate of anti-cancer immunotherapies that have moved forward through pre-clinical testing in 2D models [13,14].

Consequently, there is an urgent need to develop a multicellular tumor model that allows the incorporation of macrophages in a 3D architecture mimicking the natural TME in terms of physiologically relevant cell-cell communication, nutrient gradients, drug penetration, and hypoxic tumor regions, all being important for disease progression and drug response [15,16]. Such models allow the study of how cancer cells affect infiltrating monocytes, their acquisition of pro-tumorigenic phenotypes, and the individual effect of different cancer types. The results generated using these models will help elucidate the complex tumor-immune interactions and discover novel drug target pathways within an in vivo-like TME [17,18]. These models can potentially be used for reliable and accurate pre-clinical evaluation and screening of anti-cancer drugs targeting pro-tumoral TAMs, discarding ineffective therapies during early stages of testing, while accurately identifying drugs with high therapeutic anti-cancer potential [2,19,20]. Spheroid models are the most widely used 3D tumor model and provide a 3D setting where immune cells can migrate toward and infiltrate tumor cell clusters [14,21,22].

In this study, we present 3D multicellular tumor spheroid (MCTS) models composed of tumor cells, fibroblasts, and immune cells. We detect the infiltration of monocytes from healthy donors into spheroids generated from fibroblasts and either MCF-7 (breast cancer), HT-29 (colon cancer), PANC-1, or MIA PaCa-2 (pancreatic cancers) cancer cells. Additionally, we show how tumor spheroids can influence monocyte differentiation in vitro, as infiltrating monocytes polarize toward mature macrophages with diverse phenotypes in a cancer cell line-dependent manner. Moreover, we explore the expression of 43 soluble factors in the TME of the 3D spheroid cultures and report large differences between the cytokine profile of the different tumor spheroids before and after the addition of monocytes.

## 2. Results

### 2.1. Spheroid Formation

Cell lines from different cancer types were tested with and without fibroblasts for their ability to form applicable 3D tumor spheroids mimicking the histological structures of in vivo tumors. After 7 days of co-culture, spheroids generated from MCF-7, HT-29, or PANC-1 obtained a spherical shape and stable appearance, however, with different cell line-dependent spheroid sizes despite identical seeding conditions (Figure 1A). HT-29 cells formed tight round spheres with an even surface, whereas MCF-7 and PANC-1 spheroids appeared dense and generally rounded but with an undulating surface. MIA PaCa-2 formed large spheroids with low stability and dissociated easily even when co-cultured with fibroblasts. Overall, co-culturing of cancer cells with 1BR.3.G fibroblasts enhanced the sphericity, as spheroids were more compact and had increased stability compared to spheroids grown as monocultures. Fibroblasts did not negatively affect the overall viability of the spheroids (Figure 1B), and a 1:1 ratio of cancer cells to fibroblasts was found to generate applicable 3D spheroid structures. Histological examination identified the HT-29 spheroids to be spherical with a central core containing necrotic cells, whereas MIA PaCa-2 and PANC-1 spheroids were looser in the structure, often showing a dense cell core and loose periphery (Figure 1C). MCF-7 formed more uniformly structured spheroids with presumably polarized tumor cells.

### 2.2. Monocyte Infiltration

To introduce myeloid immune cells in our spheroid cancer models, donor-derived monocytes were added to 7-day-old cancer cell-fibroblast spheroids. Temporal analysis of infiltrating monocytes was performed by quantification of CD11b-positive cells in the dissociated spheroids by flow cytometry. CD11b^+^ cells were detected in all spheroids 3, 5, and 7 days after monocyte addition (Figure 2A). A significant difference in infiltrating CD11b^+^ cells was observed between MCF-7 and MIA PaCa-2 spheroids for all time points, indicating a cancer cell line-specific recruitment of monocytes. Generally, infiltration was highest after 3 days and then declined slowly, suggesting that the monocytes infiltrate spheroids within the first days of incubation. Before dissociation of spheroids for flow cytometry analysis, supernatants from the spheroid cultures were harvested and analyzed to detect remaining CD11b^+^ cells (Figure 2B). No CD11b^+^ cells were detected in the supernatant, indicating an absence of surviving monocytes outside the spheroids.

We then prepared paraffin-embedded cancer spheroids (PECS) and examined the presence of macrophages in spheroids by immunohistochemical staining of the pan macrophage marker CD68. We found CD68^+^ cells in MCF-7, HT-29, PANC-1, and MIA PaCa-2 spheroids at a varying frequency (Figure 3A). In HT-29 spheroids, the CD68^+^ macrophages were identified both in the necrotic core and among live cancer cells in the periphery. In MCF-7 cell spheroids, a low number of CD68^+^ cells were identified in the central areas, whereas MIA PaCa-2 and PANC-1 spheroids had CD68^+^ cells located among the loose cell structure and within the core structure of PANC-1 spheroids. Additional immunofluorescence analyses were performed on HT-29 and MCF-7 spheroids, combining staining for CD68, CD11b, vimentin, and epithelial cell adhesion molecule (EpCam) to discriminate macrophages, fibroblasts, and cancer cells (Figure 3B). The CD68^+^ macrophages in the HT-29 spheroids were generally also CD11b^+^ and were found mainly at the edge of the necrotic core. In HT-29 spheroids, the vimentin^+^ fibroblasts formed a small separate structure in the necrotic core, whereas in the MCF-7 spheroids, the vimentin^+^ fibroblasts were mingling with the cancer cells, probably creating a scaffold for the cancer cells to form structures of differentiated cells polarized toward a fibroblast-derived extracellular matrix. The CD68^+^ and CD11^+^ macrophages were identified among the fibroblasts in the core of MCF-7 spheroids. Thus, the histological examinations of spheroids confirm the data obtained by flow cytometry that monocytes infiltrate spheroids in significant numbers and mature into macrophages.

### 2.3. Study of Macrophage Differentiation in 3D Multicellular Spheroid Models

The phenotype of the infiltrated monocyte-derived macrophages (MDMs) was evaluated to explore if cancer cell-fibroblast spheroids polarize infiltrating monocytes into M2-like macrophages as often observed in vivo. The expression of the macrophage polarization markers CD14, CD40, CD86, CD163, CD206, and MHC class II markers was assessed to determine the temporal profile of macrophage differentiation in spheroids. The spheroid-polarized MDMs were compared to control M1 and M2a macrophages and unstimulated monocytes (Figure 4 and Appendix A). The surface expression profile was cancer cell line-dependent and generally increased over time, except for CD14, which only increased for MIA PaCa-2-polarized MDMs, and MHC class II, which only increased for MCF-7 polarized MDMs, indicating that the differentiation of monocytes into macrophages occurs over time. MIA PaCa-2 spheroids polarized infiltrating monocytes to M2-like macrophages with high CD206 and CD14 expression and low MHC class II and CD86 expression. On the other hand, MCF-7 spheroids gradually generated M1-like macrophages with low CD206 and CD14 expression and high MHC class II and CD86 expression. MDMs polarized by HT-29 and PANC-1 spheroids primarily obtained an M2-like phenotype with high CD206, CD163, and low MHC class II but also showed high expression of the M1 markers CD86 and CD40. These results suggest that the molecular characteristic of polarization is directed by the cancer cells and that even though monocytes had infiltrated spheroids already at day 3, full maturation to macrophages may take at least 7 days within the spheroids.

We then investigated if the monocyte infiltration and polarization were dependent on the fibroblast cell line used in the model. MCF-7, HT-29, PANC-1, and MIA PaCa-2 tumor spheroids were therefore generated using either 1BR.3.G skin fibroblasts or MRC-5 lung fibroblasts. Imaging showed no apparent differences between tumor spheroids cultured with 1BR.3.G and MRC-5 fibroblasts, apart from larger and less compact MIA PaCa-2 spheroids with MRC-5 fibroblasts (Appendix A). The number of CD11b^+^ cells was again found to be highest in MIA PaCa-2 and lowest in MCF-7 spheroids, but no difference in monocyte numbers was found between tumor spheroids cultured with 1BR.3.G compared to MRC-5 fibroblast (Appendix A). By analyzing the expression of macrophage polarization markers on the infiltrating monocytes from tumor spheroid co-cultures with 1BR.3.G or MRC-5 fibroblast, their phenotypes were found to be highly similar, suggesting that monocyte infiltration and polarization is unrelated to the fibroblast cells (Appendix A).

### 2.4. Macrophage Polarization by Tumor-Conditioned Media

We have previously generated tolerogenic dendritic cells (DCs) by tumor-conditioned media (TCM), and we hypothesized that monocytes could be polarized into TAM-like macrophages by 2D culturing in media conditioned from cancer cells [23]. TCM from MCF-7, HT-29, or MIA PaCa-2 cancer cell lines were generated and added to monocyte cultures on the day of seeding (day 0) or 6 days later (day 6) as previously optimized and described for DC TCM stimulations [23]. The effects of including the TME relevant cytokines IL-4 and IL-10 to the TCM cultures were evaluated as well. To assess the phenotype of TCM-polarized MDMs, the expression of macrophage polarization markers was analyzed and compared to control M1 and M2 macrophages and unstimulated monocytes. Generally, TCM-polarized MDMs acquired an expression profile similar to the M2 control with low expression of CD86, MHC class II, and CD40 (Figure 5 and Appendix A). The CD163 expression increased substantially compared to unstimulated monocytes not receiving TCM, especially when IL-4 and IL-10 were included in the TCM cultures. CD14 expression also increased compared to unstimulated monocytes, but the cytokines did not increase its expression further. However, upregulation of the TAM marker CD206 was only observed upon exogenous addition of IL-4 and IL-10 to TCM cultures, without any additive effect of the TCM compared to control M2 macrophages stimulated with IL-4 and IL-10 alone. This suggests that TCM is insufficient to fully polarize monocytes into TAM-like macrophages. However, individual effects of the different cancer cell lines were detected in cultures without cytokines, as significant downregulation of MHC class II and upregulation of CD206 and CD14 expression were observed between monocytes polarized by TCM from MIA PaCa-2 and MCF-7. Certain effects of the individual cancer cell lines were replicated by the TCM. For example, the increased M2-like polarization by MIA PaCa-2 and stronger M1-like polarization of MCF-7 were observed both on spheroid and TCM-polarized MDMs. These results substantiate the above assumption that distinct cytokine profiles from the various cell lines differently affect monocytes and suggest that both soluble factors and cell-cell interactions in the MCTS system play a key role in modulating the phenotype of macrophages.

### 2.5. Cytokine Profile of the Multicellular 3D Spheroid Cultures

Cancer cells, fibroblasts, and TAMs secrete cytokines and chemokines that support the survival of cancer cells and the recruitment of immune cells to the TME [8,24]. To determine the variations in cytokine secretion from the different cancer cell-fibroblast spheroids and how infiltrating monocytes affect this cytokine profile, the secretion of 43 different soluble factors including cytokines, chemokines, and growth factors was analyzed by Luminex multiplex analysis. The cytokines were measured in 14-day-old spheroid cultures with and without added monocytes and compared to control M1 and M2a macrophages and unstimulated monocytes (Figure 6A and Appendix A). As expected, spheroid cultures differed in their cytokine profile depending on the cancer cell line, both with and without MDMs. Generally, MCF-7 spheroids secreted low levels of soluble factors, whereas MIA PaCa-2 spheroids secreted high levels, possibly due to their large difference in size and viable cells. The secretion of most of the soluble factors was higher upon the addition of monocytes, in particular in HT-29, PANC-1, and MIA PaCa-2 spheroids.

Other notable observations from the analysis include that the amount of CCL2 and CCL7 was strongly increased upon the addition of monocytes to all spheroid cultures, especially in HT-29 and PANC-1 spheroids. Secretion of CCL23, -24, -25, -26, CXCL12, and -16 was increased in all spheroid cultures upon monocyte addition in a cancer cell line-dependent fashion but was only marginally secreted by control macrophages. CCL8 secretion increased upon monocyte addition to MCF-7 and HT-29 spheroids and was also high for the M1 macrophage control. Secretion of the regulatory T cell recruiting cytokine, CCL22, was also highly upregulated upon monocyte addition to all tumor spheroids. CXCL2, -5, -6, and -8 and the inflammatory cytokine MIF were heavily secreted in MIA PaCa-2 spheroid cultures only, whereas CXCL10 and CXCL11 were only secreted in HT-29 spheroid cultures and increased upon monocyte addition. The pro-angiogenic VEGF was present in HT-29, PANC-1, and MIA PaCa-2 spheroid cultures, especially in HT-29 and MIA PaCa-2, but decreased upon monocyte addition. The anti-inflammatory IL-10 cytokine was also upregulated in a cancer cell line-dependent manner upon monocyte addition to all spheroids cultures but was also present in PANC-1 and MIA PaCa-2 cultures before. Interestingly, the secretion of the anti-tumorigenic pro-inflammatory cytokine, IL-12, decreased in all spheroid cultures at different rates upon monocyte addition. Overall, these data indicate that the addition of monocytes to spheroids cultures augment a pro-tumorigenic microenvironment in the 3D cancer cell-fibroblast spheroids. 

The temporal secretion of CCL2 and CCL22 in the spheroid cultures was also measured by ELISA 3, 5, and 7 days after the addition of monocytes. All spheroids cultures with monocytes secreted high levels of CCL2 compared to unstimulated monocytes but also compared to M1 and M2a control macrophages, and the levels increased over time (Figure 6B). CCL22 was detected at all measured time points in all spheroid cultures after monocyte addition (Figure 6C). Compared to unstimulated monocytes, showing low cytokine secretion, the secretion of a variety of cytokines increased over time upon monocyte addition, indicating that the infiltrating monocytes gradually polarizes to functional macrophages within the spheroid cultures. Note that spheroid cultures with and without monocytes represent different concentrations of cells.

## 3. Discussion

It is in high demand to establish a suitable model to study the complex crosstalk between different cells in the TME, which plays an important role in tumor growth and drug responsiveness. By better recapitulating the tumor in vivo compared to 2D monolayers, 3D multicellular models may be able to provide critical insight into the role of the inflammatory TME and have emerged as an alternative to in vivo animal models [13,25]. To establish 3D models, most studies focus on one or two cell lines, but to achieve a more in vivo relevant model, more cell types may be incorporated [14]. This study demonstrates the generation of 3D MCTS models that recapitulates the interactions between three cellular TME compartments: tumor cells, fibroblasts, and TAMs. The models illustrate the infiltration of monocytes into cancer cell-fibroblast spheroids as observed in many patient tumors and their subsequent polarization into macrophages. An M2-like polarization of monocytes infiltrating pancreatic tumor spheroids and embedded in non-small cell lung cancer (NSCLC) spheroids was demonstrated by Kuen et al. and Rebelo et al. [14,16]. In our study, we have demonstrated that different tumor spheroids elicit distinct sets of secreted factors and, thus, uniquely influence monocyte infiltration and differentiation resulting in both M1, M2, and mixed macrophage phenotypes. Additionally, we were able to show that the polarization of infiltrating monocytes progresses over time and that the polarization was independent of the fibroblast cell line. Therefore, we believe that our MCTS models are highly attractive tumor models for the study of TAM infiltration and polarization, reflecting the differences between different cancer types and can be exploited as in vitro screening tools for anti-cancer compounds.

In order to study a broad spectrum of conditions, four cancer cell lines, the breast cancer MCF-7, the colon cancer HT-29, and the two pancreatic cancers PANC-1 and MIA PaCa-2 were found applicable as cancer spheroid models. These cancer types have different pathologies, and their spheroid cultures showed different qualities regarding size, compactness, and monocyte infiltration. We found that 1BR.3.G fibroblasts increased spheroid stability and the physiological relevance of the model. Kuen et al. used MRC-5 fibroblasts [16]. In our study, we compared MRC-5 fibroblasts to 1BR.3.G fibroblasts and observed the formation of smaller and possibly more compact MIA PaCa-2 spheroids co-cultured with 1BR.3.G fibroblasts. However, we still had difficulties obtaining stable MIA PaCa-2 spheroids, which seems to be a general problem with this cancer cell line [16,26]. This allowed us to conclude that the infiltration and polarization of monocytes were unrelated to the fibroblast cell line suggesting that tumor-derived factors and tumor-monocyte cell interactions are responsible for the observed monocyte infiltration and polarization. However, the use of cancer-associated fibroblasts derived from patients as used by Rebelo et al. might provide a higher degree of physiological relevance. As Kuen et al., we added monocytes to established spheroids, while Rebelo et al. included monocytes in the initial phase of spheroid formation by encapsulating them into the alginate capsules. However, this strategy does not allow the study of monocyte infiltration into spheroids as observed in in vivo tumors [14,16,27]. Of the different spheroids investigated in this study, we found the monocyte infiltration to be highest in MIA PaCa-2 and lowest in MCF-7 spheroids. High infiltration and M2 polarization in MIA PaCa-2 spheroids are in accordance with previous findings [16]. Another study investigating monocyte infiltration into different breast cancer spheroids also found poor infiltration into MCF-7 spheroids suggesting that monocyte migration into spheroids is not only dependent on cancer type but also clearly cancer cell line-dependent [28].

TAMs are often polarized in primary tumors toward an M2-like state but have also been shown to display high heterogeneity as TAMs derived from patients consist of different TAM subsets, co-express M1 and M2 markers, and depend on the tumor type, stage of tumor development, and specific location within the tumor [7,29,30,31]. The heterogeneity of TAMs was also reflected in our 3D tumor spheroid models, where the different cancer cell lines promoted variations in monocyte infiltration and macrophage polarization. In particular, MIA PaCa-2 spheroids polarized infiltrating monocytes to M2-like macrophages, whereas the MDMs polarized by MCF-7 spheroids displayed an M1-like phenotype, suggesting that in MCF-7 spheroids, macrophages do not express a pro-tumoral profile. Spheroid-polarized MDMs in HT-29 and PANC-1 primarily obtained an M2-like phenotype but also showed upregulation of M1 markers. Hence, infiltrating monocytes are gradually polarized toward mature macrophages in a cancer cell line-dependent manner. Interestingly, we found that the phenotype of spheroid-polarized MDMs was different from macrophages polarized by TCM from the same cancer cell lines. These data indicate that cell interactions between cancer cells and monocytes are important for the specific activation and polarization of macrophages and cannot be explained by the soluble tumor-derived factors only. This is in agreement with a study, which showed that only macrophages incorporated directly within tumor spheroids compared to macrophages diffusely seeded throughout the 3D collagen hydrogel displayed tumor-promoting phenotypes [32].

The main function of macrophages in the TME is tightly related to their interaction with cancer cells, resulting in the secretion of soluble factors that shape the cytokine profile of the TME [8]. In our study, we performed a thorough analysis of the cytokines in the MCTS cultures and demonstrated that the spheroids differed considerably in their cytokine profile depending on the cancer cell line, even before monocyte addition, confirming that the cancer cell lines affect infiltrating monocytes in a specific manner by their individual secretion profile. Generally, MCF-7 spheroids displayed low secretion of soluble factors. Although inflammatory breast cancers (IBC) exist, the MCF-7 cell line is considered a non-inflammatory breast cancer type, correlating with the anti-inflammatory cytokine profile, low monocyte infiltration, and subsequent M1 polarization [33]. On the other hand, IBCs have been shown to attract and differentiate monocytes into tumor-promoting, immune-suppressing M2-like macrophages [34]. On the other hand, the in vitro TME of our MIA PaCa-2 spheroid cultures displayed a very pro-inflammatory environment concurrent with the high secretion of the anti-inflammatory cytokine IL-10 and high secretion of pro-angiogenic factors such as VEGF, CXCL2, -5, -6, and -8 [35]. IL-10 is well known for its role in M2 macrophage polarization and suppression of various immune cells [36]. Inflammation has been associated as key mediators of the development and progression of pancreatic cancers, correlating with the inflammatory cytokine profile of PANC-1 and MIA PaCa-2 spheroids, the high infiltration of monocytes, and M2-like polarization, especially in MIA PaCa-2 spheroid [37].

In our experiments, the secretion of most soluble factors increased substantially upon monocyte addition to spheroid cultures. The increased secretion was generally associated with a more pro-tumoral cytokine profile, suggesting a cancer cell-specific education of infiltrating MDMs into in vivo-like TAMs, especially by MIA PaCa-2 spheroids. For instance, we detected an increase in CCL2, CCL22, and CCL24 levels in all tumor spheroids upon monocyte addition. Wang et al. have demonstrated that CCL2 is the highest upregulated chemokine gene in TAMs, and generally, CCL2 correlates with poor prognosis in cancer patients [38,39]. CCL22 and CCL24 have been associated with M2-like polarization and worse prognosis in cancers and were also found to increase upon monocyte encapsulation in NSCLC spheroids [14,40]. Additionally, we detected that secretion of IL-12, pivotal for induction of a strong antitumor immune response [41], decreased upon monocyte addition to all spheroid cultures, while IL-10 secretion increased, resulting in a more pro-tumoral and immunosuppressive TME. Moreover, high IL-10 secretion and low IL-12 secretion is a known marker for M2 macrophages, further suggesting that infiltrating MDMs generally acquire an M2-like phenotype [36]. VEGF, known to be involved in reshaping the immune microenvironment to an immunosuppressive state, was found highly expressed in HT-29, PANC-1, and MIA PaCa-2 spheroids [42]. VEGF levels have also been shown to be associated with CCL2 and IL-4 expression, important for monocyte recruitment and M2 macrophage polarization, respectively, which corresponds with our findings upon monocyte addition to spheroid cultures [43]. On the other hand, we demonstrated that monocytes infiltrating HT-29 spheroids may also possess anti-tumoral activities, as the secretion of CXCL10 and CXCL11, only secreted in HT-29 spheroids, and increasing upon monocyte addition, has been associated with tumor-suppressive functions [44,45].

Our findings by FACS analyses of identifying infiltrating donor-derived monocytes and their maturation into macrophages in cancer cell-fibroblast spheroids were strongly supported by our histological observations in PECS, as CD68^+^ macrophages were identified by immunohistochemistry in all spheroids incubated with monocytes. The histological assessment also allowed a detailed characterization of the MCTS structures. Using 4-plex immunofluorescence, we could discriminate the macrophages from fibroblasts and cancer cells and noted that fibroblasts in HT-29 spheroids form a structure in the necrotic core, whereas they directly interact with MCF-7 cells to form more well-differentiated structures.

The importance of these findings may help to further optimize spheroid formation to mimic in vivo tumor-stroma structures. We have previously shown that PECS can be used for *in situ* expression analyses for the detection of microRNA and long non-coding RNA [46]. It is tempting to speculate that the MCTS can be used not only as a model for therapeutic targeting of proteins and cytokines but also for testing anti-sense therapies targeting microRNA and other RNAs to control monocyte infiltration and macrophage maturation. Hence, our 3D MCTS models are promising tools for the screening of anti-cancer immunotherapies, but the use of permanent cancer cell lines is associated with potential limitations. Cell lines have adapted to in vitro culture conditions and lose the heterogeneity and complexity found in the original tumor, and do not resemble primary cancer cells in several parameters [47,48,49]. Therefore, the tumor spheroid models could be expanded to include primary cancer cells to be more physiologically relevant and to better investigate the cause of why patients respond differently to certain types of treatment, thereby truly personalizing cancer therapy. Furthermore, including T cells in the MCTS model could provide additional complexity allowing the study of combined immunotherapies and possibly enhanced anti-tumoral responses.

## 4. Materials and Methods

### 4.1. Spheroid Formation

A total of 5000 tumor cells were seeded per well in ultra-low-attachment 96-well plates (Corning®, Corning, NY, USA, #7007) for monoculture, and 2500 tumor cells and 2500 fibroblast cells were seeded per well for co-culture. Cells were centrifuged for 10 min at 500 G for spheroid formation and were cultured in their respective culturing media at 37 °C in 5% CO_2_ until spheroid formation. Media was changed every 3–4 days.

### 4.2. Cell Culture

MCF-7 (ATCC® HTB-22™), HT-29 (ATCC® HTB-38™), MIA PaCa-2 (ATCC® CRL-1420™), PANC-1 (ATCC® CRL-1469™), 1BR.3.G (ECACC 90020507) and MRC-5 (ATCC® CCL-171™) cell lines were originally obtained by ATTC or ECACC and maintained in Dulbecco’s Modified Eagle’s Medium (DMEM) (Lonza, Basel, Switzerland, #12-709F) containing 10% fetal bovine serum (FBS) (Biological Industries, Kibbutz Beit-Haemek, Israel, #04-007-1A), 1% Pen/Strep (Lonza, #DE17-602F), 1% sodium pyruvate (100 mM) (Gibco™, Grand Island, NY, USA, #11360070) and 1% GlutaMAX™ Supplement (Gibco™, #35050061). Cells were incubated at 37 ℃ in 5% CO_2_ and split weekly or when approaching 90% confluence. Cells used for further experiments were all below passage 20.

### 4.3. Tumor Spheroid Viability

Cell viability was measured using CellTiter-Glo® 3D Cell Viability Assay (Promega, Madison, WI, USA, #G9682) on days 7 and 14, according to the supplier’s instructions. The relative luminescence units (RLU) were measured on the Tecan infinite 200 Pro plate reader. The RLU values from samples with media only were subtracted from spheroid RLU values. Samples were run in triplicates.

### 4.4. Isolation of Human Monocytes and Generation of Spheroid-Polarized Macrophages

Buffy coats were obtained from anonymized healthy donors (Rigshospitalet, Denmark) approved for in vitro research, and PBMCs were obtained by density gradient centrifugation according to MACS Miltenyi Biotec GmbH protocol “Isolation of cells from human peripheral blood by density gradient centrifugation”. Monocytes were isolated from PBMCs by positive selection of CD14^+^ cells with CD14 microbeads, human (Miltenyi Biotec, Bergisch Gladbach, Germany, #130-050-201) according to MACS Miltenyi Biotec GmbH protocol “CD14 MicroBeads, human”. A total of 1 × 10^4^ freshly isolated human CD14^+^ monocytes were added to the cancer cell-fibroblast spheroids per well on day 7 of spheroid formation. The spheroid cultures were incubated for 3, 5, or 7 additional days without the addition of polarizing cytokines and media change. Spheroids without monocytes were included as a control.

### 4.5. In Vitro Differentiation of Control M1/M2a/M2 (IL-4+IL-10) Macrophages

Freshly isolated human CD14^+^ monocytes were seeded in 6-well plates at a concentration of 6 × 10^6^ cells per 3 mL media. Monocytes were cultured in DMEM or RPMI supplemented with 10% FBS, 1% Pen/Strep, 1% sodium pyruvate, and 1% GlutaMAX™ Supplement at 37 °C in 5% CO_2_. For M1-like polarization, the media was supplemented with GM-CSF (100 ng/mL). For M2-like polarization, the media was supplemented with M-CSF (100 ng/mL). On day 6, macrophages were M1 stimulated (20 ng/mL IFN-γ and 20 ng/mL LPS), M2a stimulated (20 ng/mL IL-4 and 20 ng/mL IL-13) or M2 (IL-4+IL-10) stimulated (20 ng/mL IL-10 and 20 ng/mL IL-4) for 24 hours. Unstimulated monocytes were cultured in media without GM-CSF or M-CSF. The appropriate medium was refreshed 3 and 6 days after seeding, and cells and supernatant were harvested on day 7. For ELISA and Luminex analyses comparing control macrophages with spheroid-polarized macrophages, the supernatant was harvested from 7-day-old cultures of M1, M2a, and unstimulated monocytes seeded in 96-well plates at an initial cell density of 1 × 10^4^ per well in 200 µL media.

### 4.6. Tumor-Conditioned Media

For the generation of tumor-condition media (TCM), 5 × 10^6^ cells were seeded into a T175 flask in RPMI 1640 (Lonza, #BE04-558F) supplemented with 10% FBS, 1% Pen/Strep, 1% sodium pyruvate (100 mM) and 1% GlutaMAX™ Supplement. After 24 hours of adaptation to the new medium, the medium was changed to a total volume of 30 mL medium per flask. After another 48 hours of incubation, the TCM was harvested and passed through a 0.45 μm filter (Merck Millipore Ltd., Carrigtwohill, Ireland, #SLHA033SS). The TCM was aliquoted and stored at −80 °C until use.

### 4.7. In Vitro Differentiation of TAM-like Macrophages by Tumor-Conditioned Media

Freshly isolated human CD14^+^ monocytes were seeded in 6-well plates (Thermo Fischer Scientific, Waltham, MA, USA, #140673) at a concentration of 6 × 10^6^ cells per 3 mL media. Monocytes were cultured in RPMI 1640 media supplemented with 10% FBS, 1% Pen/Strep, 1% sodium pyruvate, 1% GlutaMAX™ Supplement and M-CSF (100 ng/mL) and on either day 0 or day 6 after monocyte isolation, 25% TCM containing either IL-4 (20 ng/mL) and IL-10 (20 ng/mL) or no cytokines were included in the culture medium. Appropriate medium and cytokines were refreshed on days 3 and 6 after seeding, and cells were harvested on day 7.

### 4.8. Flow Cytometry

In vitro generated macrophages were detached from 6-well plates by cell scraper. To harvest infiltrated spheroid-polarized macrophages, cancer cell-fibroblast spheroids were first washed once in PBS. Spheroids were then dissociated by incubation at 37 °C with StemPro™ Accutase™ Cell Dissociation Reagent (Gibco, #A1110501) and carefully re-suspended by pipetting up and down every 10 minutes. Cells were stained for viability using the LIVE/DEAD™ Fixable Near-IR cell dye (Thermo Fisher Scientific, #L10119) and were subsequently blocked using FcR blocking reagent (Miltenyi Biotec, #130-059-901) before being stained for surface markers. As compensation controls, BD FC beads were used. For the fixable aqua-fluorescent reactive dye and fluorochromes not present in the BD FC beads, the ArC Amine Reactive Compensation Bead Kit (Invitrogen, Waltham, MA, USA) and the AbC Anti-Mouse Bead Kit (Invitrogen) were used, respectively. Flow cytometry was performed on a BD FACSLyric™ flow cytometer, and the data were analyzed with FlowLogic (Inivai Technologies, Mentone, Australia) software. For gating strategy, see Appendix A.

### 4.9. Cytokines and Reagents

Cytokines were purchased from Peprotech (Cranbury, NJ, USA): recombinant human IL-4 (#200-04), recombinant human IL-13 (#200-13), recombinant human IL-10 (#200-10), recombinant human M-CSF (#300-25), recombinant human GM-CSF (#300-03), recombinant human IFN-γ (#300-02) and Sigma-Aldrich (St. Louis, MO, USA): LPS (#L3024). Mouse anti-human antibodies were purchased from BD Biosciences (Franklin Lakes, NJ, USA CD11c FITC (#561355), CD11b FITC (#562793), CD40 BV711 (#563397), CD40 BV510 (#563456), CD80 BV480 (#562432), CD86 BV421 (#562432), MHC-II BV605 (#740408) and from Biolegend (San Diego, CA, USA): CD163 APC (#333610), CD14 PerCp Cy5.5 (#325622), CD206 PE (#321106).

### 4.10. Measurement of Secreted Chemokines over Time

The concentration of CCL22 and CCL2 in the supernatant was determined by ELISA. The supernatant was harvested from spheroid cultures with and without monocytes added on day 7. The following ELISA kits were purchased from R&D systems (Minneapolis, MN, USA): Human CCL18/PARC DuoSet ELISA (DY394), Human CCL2/MCP-1 DuoSet ELISA (DY279), and Human CCL22/MDC DuoSet ELISA (DY336). All kits were used according to the manufacturer’s instructions. Samples were run in triplicates.

### 4.11. Measurement of Cytokine Profile by Luminex

The concentration of 43 soluble factors in the supernatant was determined by Luminex multiplex technology. The supernatant was harvested at day 14 from spheroids with and without monocytes added on day 7. The following kits were purchased from Bio-Rad Laboratories, Inc. (Hercules, CA, USA): Bio-Plex Pro™ Human Chemokine Panel, 40-Plex (#171AK99MR2), Bio-Plex Pro Human IL-1ra Set (#171B5002M), Bio-Plex Pro Human IL-12 (p70) Set (#171B5011M) and Bio-Plex Pro™ Human VEGF Set (#171B5027M). All kits were used according to the manufacturer’s instructions. Median fluorescence intensity (MFI) was read on a Luminex MAGPIX® reader with XPONENT software.

### 4.12. Immunohistochemistry

Fourteen-day-old cancer cell-fibroblast spheroids were fixed and paraffin-embedded as previously described in detail to obtain paraffin-embedded cancer cell spheroids (PECS) [46]. Five µm thick sections were obtained for hematoxylin and eosin (H&E) staining and immunohistochemistry. For immunoperoxidase staining, the mouse anti-CD68 monoclonal antibody (mAb, clone KP1, Dako-Agilent, Glostrup, Denmark) was applied at 1:500 dilution after heat-induced epitope retrieval (HIER, using CC1, Roche, Basel, Switzerland) using a Ventana Discovery Ultra instrument (Roche). The mAb was detected using OmniMap Mouse and DAB chromogen. For multiplex immunofluorescence staining in the Ventana instrument, antibodies were detected with either anti-rabbit or anti-mouse OmniMap (HRP) and discovery fluorophore substrates and were applied in the following sequence: rabbit-anti-Vimentin (detected with DCC/AQUA), rabbit-anti-CD11b (rhodamin, R6G), mouse anti-CD68 (detected with Cy5), and finally rabbit-anti-EpCam (FAM). Sections were mounted with DAPI-containing mounting medium (ProLong Gold, Thermo Fisher Scientific). The rabbit-anti-Vimentin (1:4000), CD11b (1:2000), and EpCam (1:3000) antibodies were all from Abcam (Cambridge, U.K.). Antibodies were eluted from sections in between steps using the Ventana CC2 reagent (100 °C min for 8 min). Brightfield images were obtained from digital slides acquired using a Zeiss AxioScan equipped with a 20× objective (Zeiss, Oberkochen, Germany), and digital slides of fluorescence-stained sections were also obtained at 20× using a Pannoramic confocal slide scanner (3D HisTech, Budapest, Hungary) as described in detail elsewhere [50].

### 4.13. Statistics

Figures were prepared using GraphPad Prism V9.0.2 software and CorelDRAW software version 2017. Statistical significances were calculated using the Friedman non-parametric hypothesis test. Post-hoc multiple comparisons analysis using the Dunn’s test method was used to compare each sample to each other. The resulting *p*-values were automatically corrected for multiple comparisons. *p*-values less than 0.05 were considered to be statistically significant. *: *p* < 0.05, **: *p* < 0.01, and ***: *p* < 0.001.

## 5. Conclusions

The 3D multicellular tumor spheroid models established in this study allows the recapitulation of the interactions between different cell compartments of the TME and the resultant differentiation of infiltrating monocytes. Using the models, we showed that both soluble factors and cell-cell interactions in the 3D model play a key role in modulating the phenotype of macrophages since the polarization of macrophages was not promoted by exogenous cytokines or replicated by TCM in 2D cultures. We showed M2-like polarization of monocytes infiltrating MIA PaCa-2 spheroids and demonstrated that different cancer cell types elicit distinct sets of secreted factors resulting in both M1, M2, and mixed macrophage phenotypes. Hence, the different macrophage phenotypes observed in our 3D spheroid models allow the recreation of tumors from different cancer patients with different molecular characteristics, providing a model for the screening of compounds in a platform mimicking distinct in vivo situations. Consequently, expanding our multicellular tumor spheroid model to include different cancer cell lines can be used to study a variety of cancer types and their responses to therapy.

## Figures and Tables

**Figure 1 pathogens-10-00969-f001:**
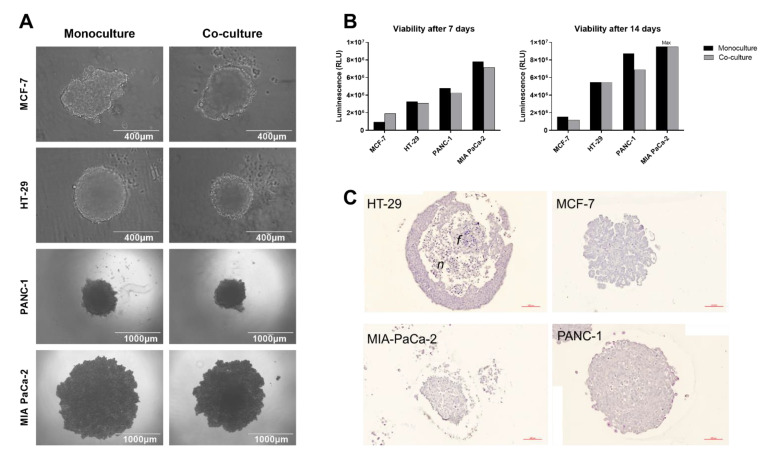
MCF-7, HT-29, MIA PaCa-2, and PANC-1 tumor cells were seeded either as monocultures or co-cultures with 1BR.3.G fibroblasts. (**A**) Brightfield images were obtained after 7 days. Brightness and contrast were adjusted for images of MCF-7 and HT-29 spheroids; (**B**) viability was measured by CellTiterGlo assays after 7 or 14 days of culture. Mean values of data from 3 spheroids from 1 individual experiment are shown; (**C**) hematoxylin and eosin staining of histological sections obtained of 14-day-old spheroid co-cultures after fixation and paraffin embedding. *n* indicates the necrotic core. *f* indicates a sub-spheroid of fibroblasts. Scale bar = 100 µm.

**Figure 2 pathogens-10-00969-f002:**
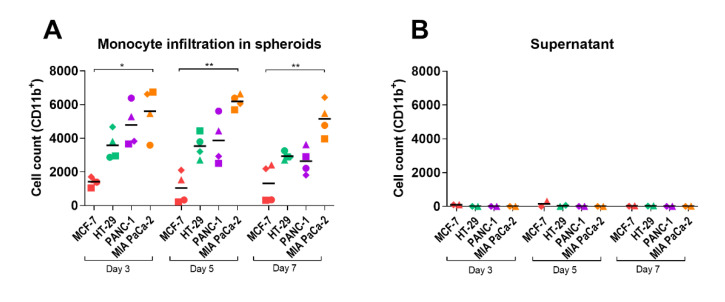
Monocyte infiltration in spheroids over time. Spheroids were generated from MCF-7, HT-29, PANC-1, and MIA PaCa-2 tumor cells and 1BR.3.G fibroblasts. CD14^+^ monocytes were added to 7-day-old spheroids, and 3, 5, and 7 days later, spheroids were analyzed by flow cytometry. (**A**) Total numbers of CD11b^+^ cells. Each point represents cells from 5 pooled and dissociated spheroids from each of 4 monocyte donors; (**B**) numbers of CD11b^+^ cells in supernatant. The supernatants from spheroid cultures were analyzed without spheroid dissociation. Each point represents supernatant from 5 pooled spheroid cultures from each of 2 monocyte donors. Statistical tests are performed within the same days. * = *p*-value of <0.05. ** = *p*-values of <0.01.

**Figure 3 pathogens-10-00969-f003:**
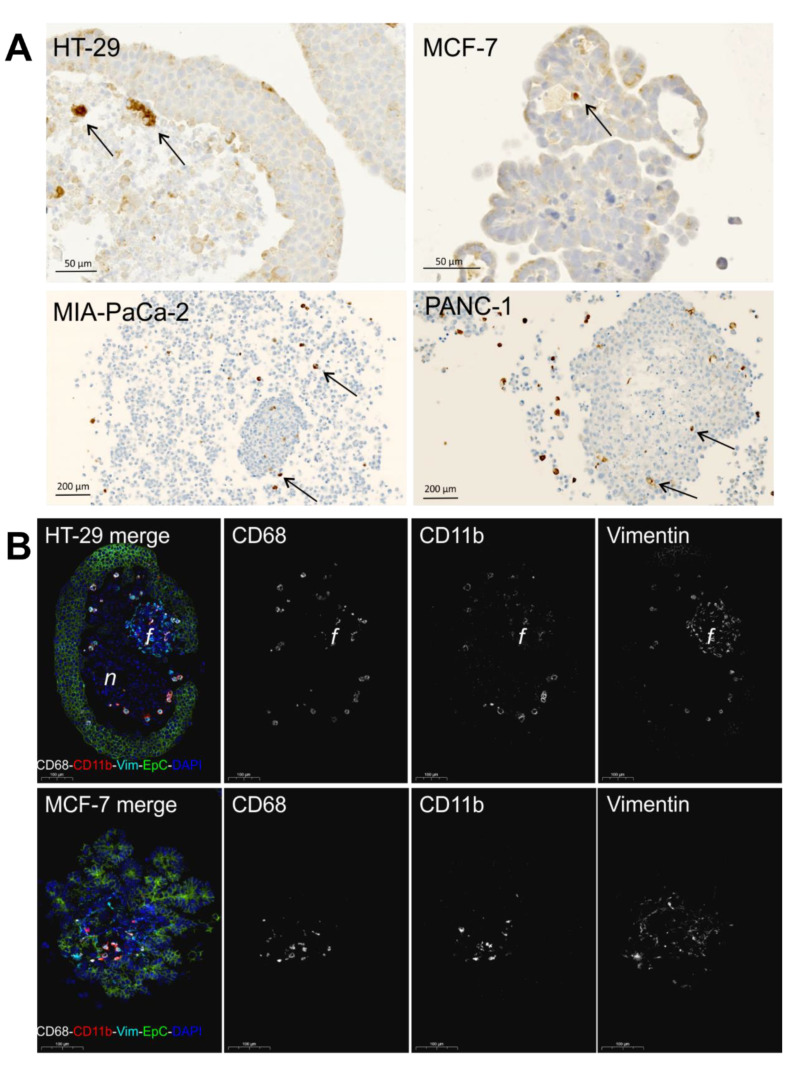
Histological assessment of macrophages in spheroids. Spheroids were generated from MCF-7, HT-29, PANC-1, and MIA PaCa-2 tumor cells and 1BR.3.G fibroblasts. CD14^+^ monocytes were added to 7-day-old spheroids, and 7 days later, spheroids were fixed and paraffin-embedded for histological analysis. (**A**) CD68^+^ macrophages by immunoperoxidase staining indicated with arrows; (**B**) CD68^+^/CD11b^+^ macrophages in MCF-7 and HT-29 spheroids by immunofluorescence staining in parallel with vimentin^+^ (Vim) fibroblasts and EpCam^+^ (EpC) tumor cells. Note that the CD68^+^/CD11b^+^ macrophages are also vimentin^+^ and that the vimentin^+^ fibroblasts form their own sub-spheroid *f* in the core of the HT-29 spheroid, whereas the vimentin^+^ fibroblasts in the MCF-7 spheroid form a mesenchymal-like scaffold for the MCF-7. *n* indicates the necrotic core. Scale bar = 100 µm.

**Figure 4 pathogens-10-00969-f004:**
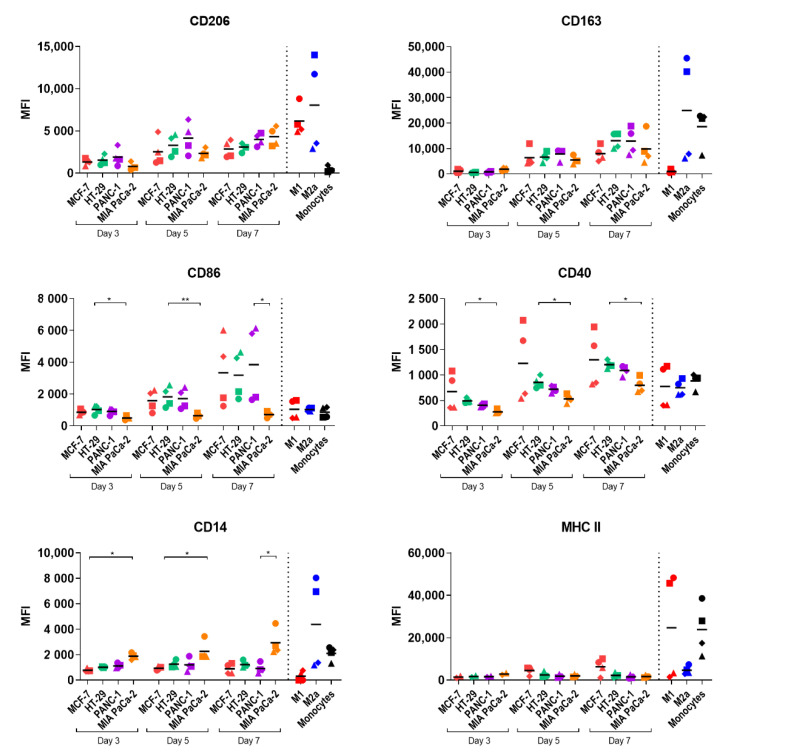
Profile of macrophages after spheroid-induced polarization over time. Spheroids were generated from MCF-7, HT-29, PANC-1, and MIA PaCa-2 tumor cells and 1BR.3.G fibroblasts. CD14^+^ monocytes were added to 7-day-old spheroids, and 3, 5, and 7 days later, spheroids were analyzed by flow cytometry. The CD11b^+^ cells were analyzed for surface expression of MHC class II, CD163, CD86, CD206, CD40, and CD14. The CD11b^+^ cells from the spheroid-polarized MDMs were compared to control M1 and M2a macrophages generated by cytokine cocktails and unstimulated monocytes. Each point represents cells from 5 pooled and dissociated spheroids from each of 4 monocyte donors, except for control M1, M2a macrophages, and unstimulated monocytes. MHC class II expression on day 3 only represents data from 1 experiment with 2 monocyte donors. Statistical tests are performed within the same days. * = *p*-value of <0.05. ** = *p*-values of <0.01.

**Figure 5 pathogens-10-00969-f005:**
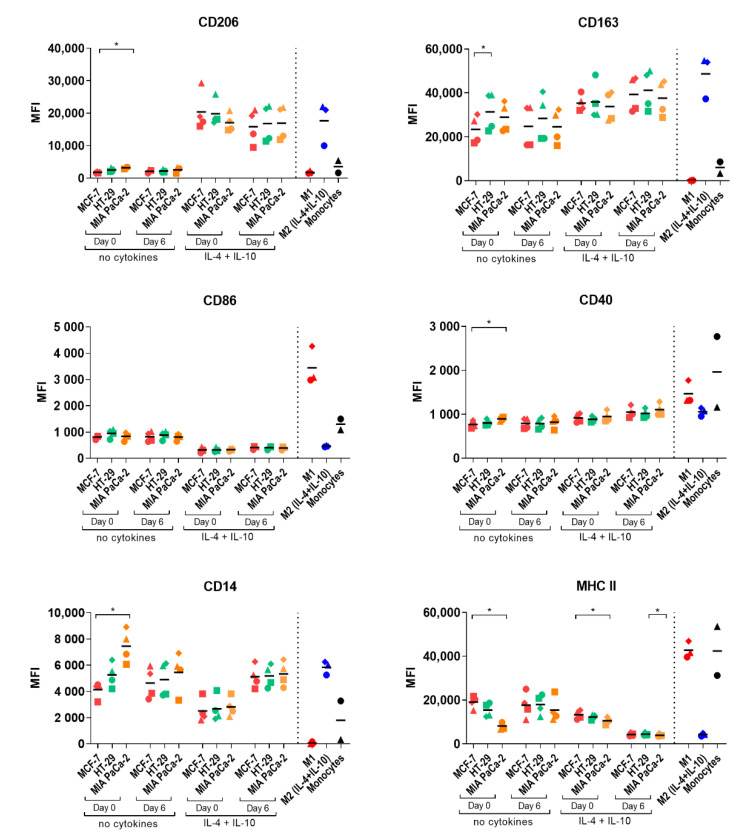
Profile of macrophages after TCM-induced polarization. On day 0 or day 6, 25% TCM from 2D cultured MCF-7, HT-29, or MIA PaCa-2 with or without cytokines (IL-4 and IL-10) was added to cultures after seeding of CD14^+^ monocytes. M-CSF is present in the media during the maturation of all TCM-polarized monocytes. After 7 days of seeding, cells were harvested and analyzed by flow cytometry. The CD11b^+^ cells were analyzed for surface expression of MHC class II, CD163, CD86, CD206, CD40, and CD14 and compared to control M1 and M2 macrophages generated by cytokine cocktails and unstimulated monocytes. Data represent 4 monocyte donors in total (M1 and M2 macrophages represent 3 donors, and unstimulated monocytes represent 2 donors). Statistical tests are performed within samples cultured in the same conditions. * = *p*-value of <0.05.

**Figure 6 pathogens-10-00969-f006:**
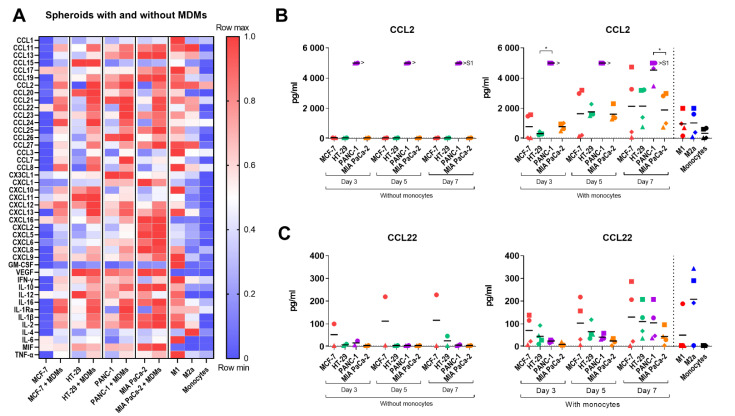
Cytokine profile of tumor spheroids with and without MDMs. Spheroids were generated from MCF-7, HT-29, PANC-1, and MIA PaCa-2 tumor cells and 1BR.3.G fibroblasts. CD14^+^ monocytes were added to 7-day-old spheroids, and cytokine levels were measured in spheroid supernatants in parallel with control M1, M2a macrophage, and unstimulated monocyte supernatants. Spheroids without monocytes were included as references. (**A**) Luminex multiplex analysis was performed on supernatants 7 days after monocyte addition. The heatmap display log-transformed min-max normalized mean values from 6 monocyte donors. Data from spheroids without monocytes represent supernatant harvested from 6 individual spheroid cultures; (**B**,**C**) CCL2 and CCL22 measured by ELISA in supernatants 3, 5, and 7 days after monocyte addition. Data represent 4 monocyte donors. Data from spheroids without monocytes represent supernatant harvested from 2 individual spheroid cultures. The dotted horizontal line represents the lower limit of detection. >S1 values represent values above the upper detection limit. Statistical tests are performed within the same days. * = *p*-value of <0.05. ** = *p*-values of <0.01.

## Data Availability

The data presented in this study are available in the article and Appendix A.

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
