# Peer review of "Monocyte Infiltration and Differentiation in 3D Multicellular Spheroid Cancer Models"

_pathogens, 2021, doi:10.3390/pathogens10080969_

Round 1
Reviewer 1 Report
The current work of Helleberg Madsen et al is an interesting contribution to the field of 3D multicellular spheroid cancer cells and monocyte infiltration. The experimental and statistical methodologies are sound.
The plagiarism check using the Turnitin tool did not reveal a significant overlap with previously published literature.
The reviewer's main issue is that cancer cell lines do not resemble primary cancer, at least at the histological level. They are used to mimic the metabolic and proliferative gradients of in vivo tumors and show clinically relevant multicellular chemoresistance. The reviewer believes that this idea needs to be thoroughly discussed.
In Peripheral blood mononuclear cells from a healthy population showed that VEGF protein levels are associated with levels of IL-4, MCP-1, and EGF (https://doi.org/10.1371/journal.pone.0220902). This goes in a similar direction to what the authors found. The authors could also discuss this.
Reviewer 2 Report
In this paper Madsen et al established an in vitro 3D multicellular spheroid model including cancer cells, fibroblasts, and monocytes to study mononuclear cellular infilration and macrophage polarisation in the TME.
The work is well-documented and the creatd in vitro modell is useful. The experiments are carried out in a rigorous and reproducible way.
Minor remarks:
- To completely cover relevant literature I would also cite the following articles in the Introduction section on macrophage polarisation and the immunosuppressive role of TAMs in the tumor microenvironment.
https://pubmed.ncbi.nlm.nih.gov/34200100/
https://pubmed.ncbi.nlm.nih.gov/30206177/
https://pubmed.ncbi.nlm.nih.gov/31940273/
https://pubmed.ncbi.nlm.nih.gov/30628894/
- Lines 71-77 in the Introduction is basically already part of the Conclusion section. At the end of the intro, it is enough to "tease" the conclusions.
- In Figure 6 the heatmap nicely illustrates cytokine profiles in a concise way. Similarly, It would be beneficial to see one or two tables summarizing the macrophage phenotypes by polarisation markers for Figure 4 and 5.
This peace of work is a nice addition to study the TME in an accessible and biologically more realistic model.
Reviewer 3 Report
Madsen et al. designed a multicellular 3D tumor model comprising of tumor cells (various tumor origin), fibroblasts and macrophages to study the interaction among cells. They found the effect of tumor cell lines on macrophage polarization. Additionally, they provided comparative analysis of 43 factors secreted by 3-D spheroids in relation to presence of macrophages.
Overall, this is an excellent study well technically done with high potential interest for scientific community. Nevertheless, some additional information should be specified to make the 3-D multicellular model designed by authors comprehensively described.
Line 88: Please, specify the type of fibroblasts (1BR.3.G) and their population doubling (PD). The information about the number/percentage of senescent cells (performed by inexpensive senescence-associated beta galactosidase test) naturally occurring in parental fibroblast culture is crucial for potential followers to be aware that senescent cells remarkably contribute to composition of TME. The early population doubling fibroblasts contain seldom senescent cells, however, their presence increases with advancing serial cultivation. Also higher age cultures change secretome, so the PD of fibroblast (including MRC-5) is important to ne indicated in Material and Methods.
For quantitative cytokine measurements, the expression taking into account of the number of producing cells (pg/ml/cell) would be more informative.
Minor issues
Please, indicate bar size for Figure 1C and Figure 3B in the legend.
I would recommend to use decimal point instead of comma in English written text (related to Figures 2, 4, 5, 6, and Table S1).
Line 233: Please, indicate the method of cytokine estimation.
Line 426: Please, keep the term "cell line" for immortalized cells only (in description of cells in Materials and Method section). In this respect, fibroblasts used in the study are not permanent cell lines. Please, include PD number for fibroblasts.
